# Osmoregulation and Physiological Response of Largemouth Bass (*Micropterus salmoides*) Juvenile to Different Salinity Stresses

**DOI:** 10.3390/ijms26083847

**Published:** 2025-04-18

**Authors:** Yang Liu, Jing Tian, Hongmei Song, Tao Zhu, Caixia Lei, Jinxing Du, Shengjie Li

**Affiliations:** 1Key Laboratory of Tropical and Subtropical Fishery Resources Application and Cultivation, Ministry of Agriculture and Rural Affairs, Pearl River Fisheries Research Institute, Chinese Academy of Fishery Sciences, Guangzhou 510380, China; 15731986210@163.com (Y.L.); zjtianjing@prfri.ac.cn (J.T.); shm1227@126.com (H.S.); zhut1009@163.com (T.Z.); leicaixia0703@sina.com (C.L.); m18621151103@163.com (J.D.); 2College of Fisheries and Life Science, Shanghai Ocean University, Shanghai 201306, China

**Keywords:** largemouth bass, salinity, osmoregulation, antioxidant enzymes, tissue damage

## Abstract

The distribution of saline-alkali water is extensive and is increasing globally each year. Fully utilizing saline-alkali water for aquaculture can help alleviate the scarcity of freshwater resources in global fisheries. As a major economic fish species, the largemouth bass (*Micropterus salmoides*) holds significant potential for aquaculture in saline-alkali water. In the present study, we evaluated its tolerance to different salinities (0 ppt, 6 ppt, 9 ppt, 12 ppt, 15 ppt, and 18 ppt) and investigated tissue pathology, serum biochemical indicators, enzyme activities of osmolality and antioxidant, and the relative expression of Na-K-2Cl 1a cotransporter (*NKCC1a*) under different saline stress (0 ppt, 6 ppt, 9 ppt, and 12 ppt). The largemouth bass 96 h mortality rate increased with increasing salinity, and the LC_50_ for 96 h was 14.28 ppt based on the mortality results. High salinity group (12 ppt) caused gill and intestinal damage, including necrosis and cell shedding, while 6 ppt had no adverse effects, and the 9 ppt between the two salinities showed an adaptive change histologically. Serum osmolality, Na^+^, Cl^−^, and cortisol levels of the high salinity group were significantly higher than of the low salinities (*p* < 0.05). Similarly, Na^+^/K^+^-ATPase (NKA), Ca^2+^-Mg^2+^-ATPase (CMA), and superoxide dismutase (SOD) activities of 12 ppt peaked at 24 h (15.7 U/mgprot, 11.5 U/mgprot, and 243 U/mgprot), which is significantly different compared to the other three groups (*p* < 0.05). The expression of *NKCC1a* was significantly upregulated at 9 ppt and 12 ppt, suggesting its role in osmoregulation. Furthermore, the expression of *NKCC1a* in the gill is 2–4 times higher than that in the intestine. These results suggested that largemouth bass can be cultured at 6 ppt and selectively bred for tolerance at 9 ppt. NKA activity, cortisol levels, and *NKCC1a* expression can be used as a marker of salinity suitability. These findings provide insight into the adaptive mechanisms underlying the physiological responses to acute salinity stress and will contribute to improving aquaculture in saline waters.

## 1. Introduction

In recent years, with the improvement of people’s consumption level, the demand for aquatic products has been increasing [1], leading to the gradual expansion of the aquaculture scale and the increasing demand for water resources [2]. However, the salinization of water is becoming increasingly serious due to a variety of factors such as environmental climate change, water conservancy project construction, and mineral resource development, coupled with the scarcity of freshwater resources due to urbanization and industrial development [3,4]; saline aquaculture has become a new trend in the aquaculture industry. Therefore, fully using saline–alkaline water for aquaculture can effectively broaden the culture space, which can not only rationally utilize natural resources but also increase production [5]. However, saline–alkaline water has a significant impact on the survival, growth, and physiological regulatory mechanisms of aquatic animals due to its complexity and diversity [6]. So far, only a few euryhaline species have been successfully cultured in saline water, such as Nile tilapia (*Oreochromis niloticus*) [7], white shrimp (*Litopenaeus vannamei*) [8], and golden pompano (*Trachinotus ovatus*) [9]. Therefore, it is important to explore the physiological responses of freshwater fishes to salinity changes and to develop potential culture strategies for aquaculture.

Salinity, as an important factor in the water environment, directly affects the osmolality balance in and out of fish [10]. When the osmolality generated by the salinity of the water environment is similar to the osmolality of the internal environment, the energy consumption used by aquatic organisms for osmoregulation is lower [11], which can increase the survival and growth rate of aquatic organisms [12]. However, if the osmolality generated by the salinity of the water environment is significantly higher or lower than that of the internal environment, aquatic organisms are unable to regulate the balance between their internal and external osmolality, resulting in uncontrolled changes in the internal environment ions, which affects normal physiological activities, and in severe cases can lead to death [13]. At present, osmoregulation is mainly linked to osmotic enzymes and hormones in response to salinity stress. The corresponding Na^+^/K^+^-ATPase (NKA) and Ca^2+^-Mg^2+^-ATPase (CMA) activities of white shrimp were elevated under salinity stress [14] and rainbow trout (*Oncorhynchus mykiss*) [15]. Cortisol concentration in black seabream (*Acanthopagrus schlegelii*) increased with increasing salinity when undergoing low-salinity stress, and changes in cortisol concentrations in fish are related to ionic regulation and the mobilization of energetic substrates, as shown in a previous study [16]. The oxidative defense system was significantly affected by salinity changes. It has been found that the activities of superoxide dismutase (SOD), catalase (CAT), and glutathione peroxidase (GPX) were elevated in Nile tilapia when exposed to salinity stress [17]. Ding [18] found that salinity stress in fish leads to the production of reactive oxygen species (ROS) that induce oxidative stress, while with the inhibition of CAT and SOD, the antioxidant capacity of fish was usually reduced. In summary, previous research on fish adapting to salinity changes predominately focused on osmoregulation and changes in antioxidant capacity.

Largemouth bass (*Micropterus salmoides*), commonly known as California bass, is a high-quality freshwater fish with the advantages of strong adaptability, fast growth, easy catching, and shorter culture cycle. It is popular in the market and with consumers, which has resulted in it becoming an important economic fish in China’s freshwater aquaculture industry [19]. In northwestern China, saline intrusion has stalled largemouth bass production despite market demand [20]. In addition, current studies have shown that the largemouth bass experienced a decrease in feed intake and a decrease in specific growth rate of about 28% at salinity above 8 ppt [21]. Although several studies have investigated salinity effects on intestinal microbiota and muscle texture in largemouth bass [22], identified salinity-related growth SNPs and candidate genes [23], and characterized gene expression under short-term salt stress [24], the osmoregulation and antioxidant responses of largemouth bass under salinity stress remain unclear. To better understand how the largemouth bass cope with salinity stress, the present study investigated the effects of different salinity stress on mortality, histology, serum biochemical indexes, osmoregulation, antioxidant capacity, and the gene expression of largemouth bass. This work would provide new insights into the saltwater culture and selective breeding of salinity tolerance of largemouth bass.

## 2. Results

### 2.1. Determination of Median-Lethal Concentration

During acute exposure, the mortality rate of largemouth bass increased with the increasing salinity within 96 h, and mortality rates in 12 ppt, 15 ppt, and 18 ppt groups were 5%, 80%, and 100%, respectively (Figure 1A). However, 0 ppt, 6 ppt, and 9 ppt groups did not significantly affect the survival rate of largemouth bass. According to the nonlinear regression analysis, the 96 h—LC_50_ was 14.28 ppt (Figure 1B).

### 2.2. Histological Analysis of Gill and Intestine After 96 h Acute Salinity Exposure

The effect of increased salinity on the gill and intestine tissues of largemouth bass was significant. The gill tissue structure of 0 ppt and 6 ppt groups was complete, and chlorine cell size was normal. However, chlorine cells of 9 ppt gradually increased and expanded, which increased by 53% and 44% compared with the control groups (Figure 2). Furthermore, the gill lamella length of 9 ppt was shortened by about 25% compared to the control group. However, the quantity and diameter of chlorine cells decreased significantly at 12 ppt compared to the control group (*p* < 0.05), while the gill lamella length was reduced by 50% (Table 1). Similarly, the intestine of 0 ppt and 6 ppt groups showed normal goblet cells, intact intestinal wall, and intestinal villi. The intestinal villous epithelium of 9 ppt group exhibited an irregular morphology, and the number of goblet cells was significantly increased compared with the control group (*p* < 0.05). However, the quantity of goblet cells was significantly reduced compared with the control group (*p* < 0.05). Furthermore, the intestinal mucosal layer and intestinal villi were severely denatured and shed, resulting in significant damage in the 12 ppt group (Figure 3).

### 2.3. Effects of Different Salinity on Osmolality and Serum Ions

The serum concentrations of Na^+^, Ca^2+^, and Cl^−^ exhibited a similar trend to osmolality after the challenge, gradually increasing from 12 to 72 h and showing a proportional relationship with salinity (Figure 4A–D). Specifically, the Na^+^ concentration, Cl^−^ concentration, and serum osmolality were significantly higher in the 9 ppt and 12 ppt groups than in the 6 ppt and control groups (*p* < 0.05). However, a different trend was observed with K^+^ and Mg^2+^ (Figure 4E,F). Serum K^+^ concentration increased sharply and reached the peak value at 12 h, the concentration of each group decreased to the 1/4 level of 12 h at 24 h. Subsequently, the serum K^+^ concentration in the 12 ppt group rose sharply, which was significantly higher than that of the other three groups at 72 h and 96 h (*p* < 0.05). In addition, Na^+^ and Cl^−^ are major components of serum ions, Ca^2+^, Mg^2+^, and K^+^ possessed lower concentrations.

### 2.4. Serum Cortisol Concentration Increased with Salinity Increasing

The serum cortisol concentrations in the 6 ppt and 9 ppt groups exhibited a continued increase, with significantly higher levels observed in the former group compared to the latter during 48 to 72 h (*p* < 0.05). The serum cortisol concentration in the 12 ppt group showed a sustained rise from 0 to 48 h and decreased from 72 to 96 h, which was significantly higher than the other three groups within 48 h (Figure 5).

### 2.5. Enzyme Activity in Gill and Liver During 96 h Acute Salinity Exposure

The activity of NKA and SOD showed a trend of first increasing and then decreasing as salinity increased under different salinity stress at 96 h, and the peak value of each salinity group appeared at 24 h (Figure 6A,C). The activity of CMA and CAT exhibited a pattern of initial increase, followed by a decrease, and then increase again (Figure 6B,D). These activities rose from 0 to 24 h and were significantly higher in the 12 ppt group compared to the other three groups (*p* < 0.05). After 24 h, the activity levels in both the 9 ppt and 12 ppt groups were higher than those at 48 h. However, there was a significant difference in the activities of CMA and CAT between these two groups in 96 h: the CMA activity in the 9 ppt group was significantly higher than that in the 12 ppt group, whereas the CAT activity in the 12 ppt group was significantly higher than that in the 9 ppt group. It might be that CMA was limited due to the injury to the gill tissue in the later period, while CAT may activate its function again due to the increase in ROS in the body.

### 2.6. Effect of Different Salinity on NKCC1a Gene Expression

The predominant role of Na-K-2Cl 1a (*NKCC1a*) was observed in the gill and intestine, with higher expression levels than other tissues [24]. Therefore, the gill and intestine were selected for analyzing *NKCC1a* expression. In both the gill and intestine, the expression of *NKCC1a* exhibited an initial increase followed by a decrease (Figure 7). The expression level of *NKCC1a* in the 12 ppt group was significantly higher than in the other three groups from 12 to 48 h (*p* < 0.05), but it was significantly lower than that at 72 to 96 h (*p* < 0.05), possibly due to tissue damage caused by prolonged exposure to high salinity stress. In addition, the expression of *NKCC1a* was higher in the gill than in the intestine, indicating its tissue-specific expression profile.

## 3. Discussion

Salinity is an important abiotic environmental factor in aquaculture and impacts the growth and physiological activities of fish and other aquatic animals [25]. Largemouth bass is a potential fish for saltwater culturing due to its wide adaptability [26]. Mortality is a direct indicator of whether fish are salt tolerant, and the key factors determining salt tolerance are osmoregulation and antioxidant capacity [27].

In this study, the mortality rate of largemouth bass increased with increasing salinity, and the 96 h—LC_50_ of salinity was 14.28 ppt, similar to previous studies. For instance, some carp and perch will be severely affected or even dead in 12 ppt to 15 ppt water. The survival rate of mandarin fish (*Siniperca chuatsi*) decreased dramatically in the 14 ppt salinity; the mortality rate was 100% in 16 ppt salinity [28]. Grass carp (*Ctenopharyngodon Idella*) had varying degrees of mortality in 12 ppt, 16 ppt, and 20 ppt groups, with a mortality rate of 83.33% in 12 ppt and 100% within 24 h in both the 16 ppt and 20 ppt groups [29]. Therefore, there was a suitable range for the largemouth bass culture, and it is of great significance to explore the suitable salinity.

The gill and intestine are important organs for osmoregulation in bony fishes, whose morphological structure can undergo significant changes during the adaptation to complex external environments. The gill facilitates ion exchange between the body and the water environment through osmosis to maintain ion balance, while the intestine efficiently absorbs water to compensate for dehydration caused by the hypertonic environment [30]. Consequently, the effects of high salt stress on fish were typically assessed through histopathological analysis of gill and intestinal tissues [31]. In this study, the 9 ppt group exhibited increased and expanded chloride cells, along with shorter and wider gill filaments compared with the control group. Similarly, the quantity of goblet cells in intestinal tissue increased compared with control group. However, the gill and intestinal tissue were severely damaged at 12 ppt, and the quantity of chlorine cells and goblet cells was also drastically reduced. In previous studies, amur minnow *(Phoxinus lagowskii*) [32] and african catfish (*Clarias gariepinus*) [33] in response to salinity stress have shown that with increasing salinity, gill filaments gradually denatured and fell off, the number of chloride cells increased, and intestinal tissue also exhibited damage and loss of goblet cells and intestinal villi. It can be inferred that the proliferation of chlorine cells in the gills and goblet cells in the intestine is an adaptive response to salinity stress, while the damage of gill patches and intestinal villi is a sign of body collapse. Interestingly, the irregularity of the 9 ppt intestinal villi margins has not been reported in previous studies, which, combined with other data from this study, may be another manifestation of the body’s adaptation to salinity. In summary, the chlorine cells in the gill and goblet cells in the intestine would be increased when largemouth bass adapted to salinity stress, and the tissues would deform and fall off when the salinity is too high (12 ppt).

When fish enter a high-salinity environment, the water in the body will be lost to the outside due to the difference in the osmotic pressure inside and outside the body. Consequently, osmoregulation was indispensable in fish [34]. The water–salt balance in the osmoregulation of bony fish primarily involves the movement of Na^+^, K^+^, Cl^−^, Ca^2+^, and Mg^2+^ ions, which are regulated by two key enzymes (NKA and CMA), whose activities were often correlated with osmolality trends. The principal function of the NKA enzyme was to transport Na^+^ into the cell and K^+^ out of the cell [35]. In this study, NKA activity exhibited an initial increase in response to salinity stress but subsequently decreased in later stages due to gill tissue damage caused by prolonged exposure to high salinity (Figure 4A). Serum Na^+^ and Cl^−^ concentrations in largemouth bass gradually increased over time between 0 and 72 h, which is associated with salinity levels. The concentration of K^+^ increased abruptly at 12 h, then decreased before increasing again at 72 h. The results indicated that K^+^ entered the serum by passive diffusion at 12 h, after which the NKA enzyme facilitated the transport of Na^+^ and Cl^−^ into and out of the serum, leading to a decrease in K^+^ concentration between 24 and 72 h. The decreased NKA activity in later stages resulted in the passive diffusion of K^+^ back into the serum, causing an increase between 72 and 96 h (Figure 4E). The biphasic pattern of osmoregulation enzyme activity characterized by initial activation followed by a decline aligns with observations in spotted sea bass (*Lateolabrax maculatus*) during seawater acclimation [36], indicating conserved regulatory strategies for salinity adaptation. Temporal fluctuations in serum K^+^ levels, mediated by passive diffusion and active transport, further validate that NKA dysfunction induces K^+^ dysregulation, which is supported by studies in grass carp [29]. The CMA enzyme, another key indicator of osmoregulation, transports Ca^2+^ out of the cell and Mg^2+^ into the cell [37]. In this study, CMA activity increased in the early stages, leading to a decrease in Ca^2+^concentration and an increase in Mg^2+^ concentration. An upward trend in CMA activity with increasing salinity was similarly observed in rainbow trout during salinity stress experiments [15]. However, gill tissue damage in later stages led to a decrease in CMA activity, with a tendency to increase Ca^2+^ concentrations and decrease Mg^2+^ concentrations. In conclusion, the alterations in serum ion composition were consistent with the principles of osmosis. Short-term salinity stress leads to increased NKA and CMA activity, maintaining in vitro and in vivo osmolality in balance, but prolonged exposure to high salinity stress can result in tissue damage, culminating in decreased enzyme activity.

In addition, some hormones play an important role in osmoregulation in fish, which can control the homeostasis of osmolality by adjusting the amount of salt in the body [38]. Cortisol is a corticosteroid involved in regulating osmoregulation and metabolism in fish [39], which stimulates ion-to-ion exchange by downregulating the expression of nitric oxide synthase (NOS) and blocking the inhibition of Na^+^/K^+^-ATPase (NKA) by NOS [40]. In this study, cortisol concentrations exhibited an upward trend from 0 to 72 h with increasing salinity and stress duration, indicating that cortisol actively participated in the regulation of osmolality. Moreover, the cortisol concentration in 12 ppt was significantly higher than that of the other three groups (*p* < 0.05), indicating that the release of cortisol was promoted under high salinity stress. Previous studies found that cortisol levels in carp raised under high-salinity environments were higher than those in freshwater groups. A study reported by Bu [41] also found that cortisol concentrations were higher in tilapia raised in a high-salinity environment compared with a low-salinity. The above results showed that the level of cortisol secretion increased with the increasing salinity, which could be used as one of the positive evaluation indicators for fish facing salinity stimulation.

In addition, antioxidant capacity is another major factor in fish adaptation to salinity stress. SOD and CAT were the first line of defense against cellular oxidation and are important components of the fish antioxidant system [42]. In this study, SOD and CAT activities increased at 12 and 24 h under salinity stress, indicating that salinity stress activated the antioxidant system in fish, with a more pronounced response at higher salinities. It has been found that when salinity changes, the activity of SOD in fish increases, increasing the amount of H_2_O_2_, which leads to an increase in the reactivity of the H_2_O_2_-scavenging enzyme CAT [43]. In tilapia and Heilongjiang Minnows (*Rhynchocypris lagowskii*), high salinity stress also increased SOD activity and CAT activity [7]. However, the SOD and CAT activities in 6 ppt, 9 ppt, and 12 ppt decreased after 48 h of salinity stress (Figure 6), suggesting that the late adaptation to the salinity stress of 6 ppt and 9 ppt reduced the antioxidant response in vivo, while the salinity of 12 ppt exceeded the tolerance range and led to a decrease in enzyme activity. This is similar to the study of Chinese mitten crab (*Eriocheir sinensis*) in response to high alkali stress [44]. In conclusion, the antioxidant system revealed the limit of salinity adaptation. Early SOD and CAT activation conform to a “stimulus-effect” model, in which moderate stress enhances antioxidant defenses [34]. Subsequently, the enzyme declined at 48 h in the 12 ppt group reflected the exhaustion of the oxidative system, delineating the upper limit of salinity tolerance for this species. Notably, the sustained antioxidant activity in the 6–9 ppt group contrasted sharply with the surge in antioxidant enzyme activity observed in guppy (*Poecilia reticulata*) with high salt exposure [45], suggesting an evolutionary adaptation of largemouth bass to periodic saltwater exposure.

Fish typically expend additional energy to maintain a dynamic balance of various ions and water inside and outside the cell when salinity increases and decreases. There are usually two coping strategies: one is to change the structure, number, and type of chloride cells in the gill epithelium, and the other is to increase the expression levels of genes with ion transport activity, such as the Na-K-2Cl cotransporter (*NKCC1a*) [46]. NKCCs are a class of electroneutral transmembrane transporters that transport ions across membranes at a ratio of 1 Na^+^:1 K^+^:2 C1^−^ and include two genotypes, *NKCC1* and *NKCC2* [47]. In this study, the expression level of *NKCC1a* in the gill and intestine increased over time between 0 and 48 h in response to salinity stress, and the expression level was significantly higher in the 12 ppt salinity group than in the other three groups (*p* < 0.05) (Figure 7). The expression level of *NKCC1a* in Mozambique tilapia was also significantly higher in high salinity than in freshwater [24]. The results indicated that salinity stress induced the expression of *NKCC1a*, improved the irons transport, and promoted the regulation of osmolality. In addition, the expression level of *NKCC1a* in the gill was consistently higher than that in the intestine, indicating its primarily functions in gill. It has been found that European eel (*Anguilla japonica*) and Mozambique tilapia *NKCC1a* was expressed in most tissues, especially in the gill, which may be closely related to the salt secretion activity of chloride cells in gill filaments [48]. The above data further refine this model by demonstrating that largemouth bass prioritizes branchial over intestinal *NKCC1a* activation even at moderate salinities (12 ppt), a strategy distinct from the gut-centric osmoregulation observed in hypersaline-adapted killifish (*Fundulus heteroclitus*) [22]. In conclusion, salinity stress increased the expression level of *NKCC1a* in largemouth bass, and it primarily played a role in the gill.

## 4. Materials and Methods

### 4.1. Experimental Animals

Healthy largemouth bass were obtained from Guangdong Liangshi Aquatic Seed Industry Co., Ltd. (Foshan, China). The fish were transferred to the laboratory and temporarily reared in a non-circulating system of concrete tanks for 7 days. During the acclimation period, the fish were fed twice daily with a commercial floating feed. The temperature, pH, and dissolved oxygen were 23.4 ± 1.3 °C, 7.5 ± 0.5, and 6.50 ± 0.30 mg/L, respectively. The total ammonia concentration never exceeded 0.20 mg/L. After acclimation, healthy individuals weighing 20.3 ± 1.3 g were selected for experimentation.

### 4.2. Experimental Design and Sample Collection

The experiment included two parts: the first one was used to test the mortality rate at different salinities and calculate the lethal concentration 50 (LC_50_) for 96 h, and the second experiment was used to analyze its physiological and biochemical response.

Experiment 1: The salinity of the water was prepared using marine salt (Jiangxi Yantong Technology Co., Ltd., Ji’an, China), with salinity verified daily using a calibrated salt meter (ATAGO, Tokyo, Japan). Six salinity groups including 0, 6, 9, 12, 15, and 18 ppt were chosen, and each contained 50 individuals. The salinity was increased stepwise: starting at 0 ppt, increasing salinity by 3 ppt per hour until reaching the target salinity. Water quality parameters (temperature, pH, DO) were monitored hourly [28]. We used 96 h as the standard duration for assessing the lethal effects of salinity stressors on fish according to the Guidelines for Aquatic Toxicity Testing [49]. Fish were considered dead if they were immobile and showed no response to contact with a glass rod. Dead fish were removed from the tanks immediately. Mortalities were recorded at 12, 24, 48, 72, and 96 h of exposure, and the lethal concentration (96 h—LC_50_ values with 95% confidence level) of salinity was determined. The 96 h—LC_50_ values with 95% confidence intervals were determined using probit analysis in SPSS 24.0 (IBM, Amunk, NY, USA).

Experiment 2: The salinity levels of 0, 6, 9, and 12 ppt were selected for the 96 h salinity stress experiment based on the mortality results of experiment 1. The salt-raising method was mentioned above. After 0, 12, 24, 48, 72, and 96 h of exposure, six fish from each group were randomly sampled and euthanized using MS—222 (0.5 g/L). A total of 200 μL of blood was drawn from the tail vein using a sterile syringe and the serum was obtained by centrifugation (4000 r min^−1^, 4 °C for 10 min) after 4 h at 4 °C. At the same time, the liver, intestine, and gill tissues (second branchial arch) were quickly collected in liquid nitrogen and transferred to a −80 °C freezer for storage.

### 4.3. Histological Analysis of Gill and Intestine

It was found that the gill and intestine are important tissues for osmoregulation in teleost fishes, and their morphological structure and molecular mechanism can undergo significant changes in the process of adapting to the complex external environment [50]. Therefore, histological observations were performed to analyze whether different salinities caused damage to the gill and intestine after 96 h exposure. Intestine and gill tissue fixed in Bouin’s fixative was dehydrated in ethanol solutions and examined using standard paraffin histology procedures. Tissue blocks were cut into 5 μm sections and stained with hematoxylin and eosin [51]. Finally, the slides were mounted with neutral gum, and images were analyzed under an optical microscope (Nikon Eclipse E100, Tokyo, Japan).

### 4.4. Analysis of Serum Indicators

Changes in serum ion composition and osmolality directly reflected changes in osmoregulation in fish. In addition, some hormones such as cortisol (COR) are highly sensitive to the salinity environment to a certain extent, which can control the osmotic homeostasis in the body by regulating the salt content of fish [52]. The ion concentrations (Na^+^, K^+^, Ca^2+^, Mg^2+^, and Cl^−^) and COR in the serum were measured to better explore the mechanism of osmoregulation. Serum osmolality was determined using a BS-100 freezing point osmometer (Shanghai Yida Medical Equipment Co., Ltd., Shanghai, China). The content of Na^+^, K^+^, Ca^2+^, Mg^2+^, and Cl^−^ were all tested using a commercially available kit (Nanjing Jiancheng Institute of Bioengineering, Nanjing, China). Fish serum COR was tested using an ELISA scientific research kit (Jiangsu Enzyme Free Industry Co., Ltd., Wuxi, China), and the specific operation was carried out according to the instructions. The absorbance of these assays was measured by the Biotek Cytation5 Multimode Microplate Reader (Santa Clara, CA, USA).

### 4.5. Enzyme Activity Analysis

Osmotic enzymes (NKA and CMA) in gill and antioxidant enzymes (SOD and CAT) in the liver play an important role in responding to salinity stress in fish. Therefore, its variation was detected in salinity stress. Homogenate preparation for determining the activity of various enzymes was performed as previously described by Wang [53]. The NKA, CMA, SOD, and CAT were determined using a commercial assay kit (Nanjing Jiancheng Bioengineering Institute, Nanjing, China), respectively. The measurement method was carried out according to the instructions.

### 4.6. Total RNA Extraction, Reverse Transcription, and Real-Time Quantitative PCR Assay

Real-time quantitative PCR (qPCR) was used to quantify *NKCC1a* in the intestine and gill of exposed fish. Total RNA was extracted from six gill and intestine tissues in each group using Trizol reagent (CWBIO, Beijng, China). RNA samples were tested for integrity and concentration using 1% agarose gel electrophoresis and a Cytation5 Multimode Microplate Reader. First-strand cDNA was synthesized using the PrimeScripTM RT reagent kit with the gDNA Erase kit (Takara, Beijing, China). qPCR was performed with LightCycler96. A 5-fold dilution of cDNA was used as a template for the qPCR reaction. The reaction procedure was 95 °C for 30 s, 94 °C for 5 s, 60 °C for 30 s, and 72 °C for 30 s for 40 cycles; the dissolution curves were 95 °C for 5s, 60 °C for 1min, and 95 °C for 1 s. β-actin was used as the internal reference gene, and the relative gene expression was calculated by the 2^−ΔΔCt^ method [54]. The details of the gene primer sequences are shown in Table 2, and all primers were synthesized by Shanghai Sangon Biotech Company (Shanghai, China).

### 4.7. Statistical Analysis

All data are expressed as mean ± standard error (SE). The normality of data distributions within each group was evaluated using the Shapiro–Wilk test, and the homogeneity of variances was verified by Levene’s test. One-way ANOVA followed by Tukey’s honestly significant difference (HSD) post hoc test was performed to compare differences among groups, with statistical significance defined as *p* < 0.05. All analyses were conducted using SPSS 26.0 (IBM, USA) for hypothesis testing and GraphPad Prism 9.0 for data visualization.

## 5. Conclusions

This study investigated the dynamics of mortality, osmoregulation, and antioxidant responses in largemouth bass under different salinity stresses. This finding revealed some key advances in euryhaline adaptation mechanisms. First, the largemouth bass has been found to survive normally and adapt at salinities below 9 ppt; death and tissue damage occur when salinity exceeds 12 ppt. Second, cortisol concentrations were positively correlated with salinity and stress duration, which could serve as one of the evaluation indicators under salinity stress. Third, the activities of osmotic enzymes (NKA and CMA) and antioxidant enzymes (SOD and CAT) in largemouth bass increased rapidly within 24 h, indicating that largemouth bass can regulate osmolality and activate the antioxidant system in a short period. Furthermore, salinity stress promoted the expression of *NKCC1a*, which exhibited a tissue-specific expression pattern in the gill and intestine. In conclusion, largemouth bass can serve as a potential cultured species under suitable salinity, cortisol concentration, NKA enzyme concentration, and *NKCC1a* expression could be used as molecular markers for the salinity adaptation process of largemouth bass. These findings of our study can provide a reference for the culture of largemouth bass in saline–alkaline water.

## Figures and Tables

**Figure 1 ijms-26-03847-f001:**
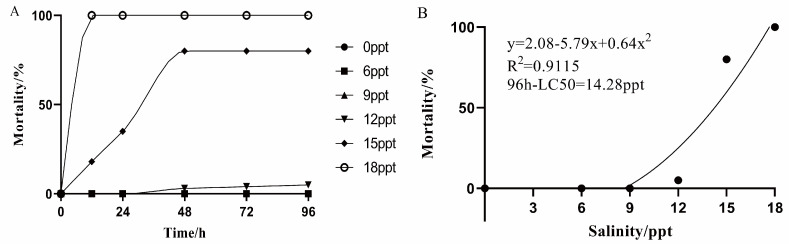
Mortality (**A**) and nonlinear regression analysis (**B**) of largemouth bass within 96 h under different salinity stresses.

**Figure 2 ijms-26-03847-f002:**
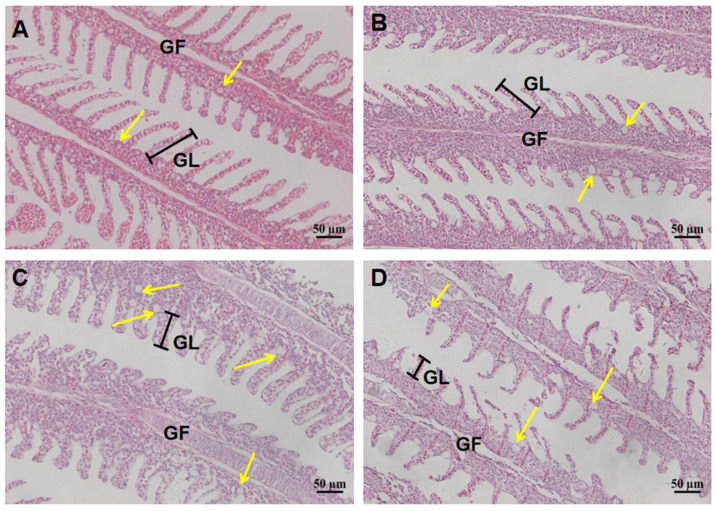
Gill histological changes at different salinity stress for 96 h. (**A**) 0 ppt; (**B**) 6 ppt; (**C**) 9 ppt; (**D**) 12 ppt. The yellow arrows point to chlorine cells. Gill filament and gill lamella length abbreviated as GF and GL, respectively.

**Figure 3 ijms-26-03847-f003:**
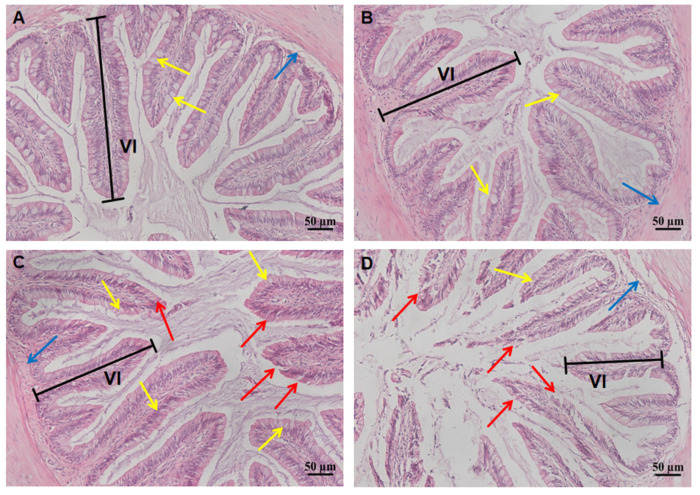
Intestinal histological changes at different salinity stress for 96 h. (**A**) 0 ppt; (**B**) 6 ppt; (**C**) 9 ppt; (**D**) 12 ppt. The yellow arrows point to goblet cell, blue arrows point to submucosa, and red arrows point to intestinal villus appearing to be injury. Intestinal villi is abbreviated as VI.

**Figure 4 ijms-26-03847-f004:**
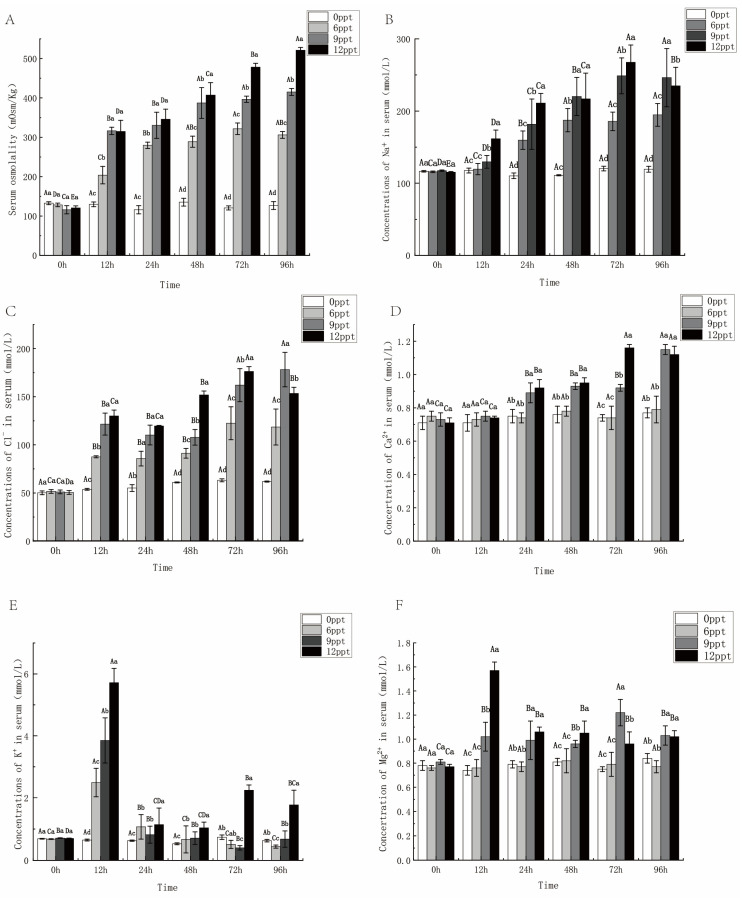
Serum ion concentration and serum osmolality within 96 h under different salinity stresses. Serum osmolality (**A**), serum Na^+^ concentration (**B**), serum Cl^−^ concentration (**C**), serum Ca^2+^ concentration (**D**), serum K^+^ concentration (**E**), and serum Mg^2+^ concentration (**F**). Data are presented as mean ± SE (*n* = 6). Different uppercase letters indicate significant differences between the same salinity at various times, and different lowercase letters indicate significant differences between different concentrations at the same time, *p* < 0.05.

**Figure 5 ijms-26-03847-f005:**
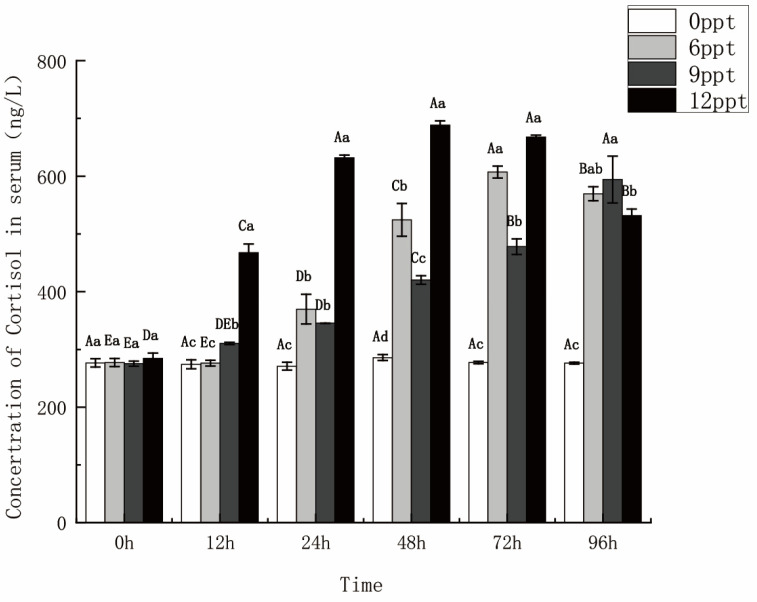
Serum cortisol concentrations within 96 h under different salinity stresses. Data are presented as mean ± SE (*n* = 6). Different uppercase letters indicate significant differences between the same salinity at various times, and different lowercase letters indicate significant differences between different concentrations at the same time, *p* < 0.05.

**Figure 6 ijms-26-03847-f006:**
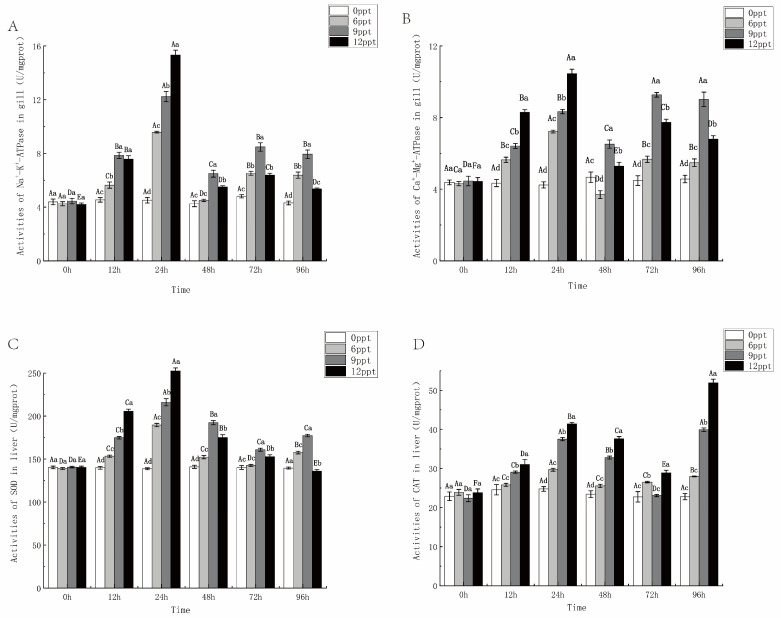
The activities of NKA (**A**) and CMA (**B**) in the gill and the activities of SOD (**C**) and CAT (**D**) in the liver within 96 h under different salinity stresses. Data are presented as mean ± SE (*n* = 6). Different uppercase letters indicate significant differences between the same salinity at various times, and different lowercase letters indicate significant differences between different concentrations at the same time, *p* < 0.05.

**Figure 7 ijms-26-03847-f007:**
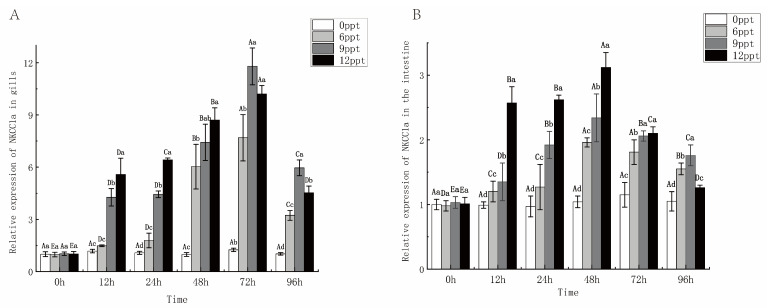
Relative expression of *NKCC1a* in the gill (**A**) and intestine (**B**). Data are presented as mean ± SE (*n* = 6). Different uppercase letters indicate significant differences between the same salinity at various times, and different lowercase letters indicate significant differences between different concentrations at the same time, *p* < 0.05.

**Table 1 ijms-26-03847-t001:** Changes in gill and intestinal histological indexes.

Indexes	0 ppt	6 ppt	9 ppt	12 ppt
Chlorine cells diameter/µm	5.3 ± 0.9 ^b^	5.6±0.7 ^b^	8.6 ± 1.3 ^a^	3.1 ± 1.1 ^c^
Chlorine cells quantity	8 ± 1.5 ^b^	8 ± 1 ^b^	11.5 ± 2.2 ^a^	5 ± 0.5 ^c^
Gill filament length/µm	31.7 ± 3.1 ^a^	29.3 ± 2.6 ^a^	23.5±2.4 ^b^	15.7 ± 1.1 ^c^
Goblet cells quantity	22.3 ± 1.5 ^b^	23.1 ± 1.8 ^b^	35.6 ± 3.1 ^a^	19.8 ± 3.7 ^c^

Chlorine cells are calculated as the number of gill filaments per unit length of 50 µm, and goblet cells are calculated as the number of intestinal villi per unit length of 50 µm. Data are presented as mean ± SE (*n* = 6); different lowercase letters indicate significant differences between different concentration (*p* < 0.05).

**Table 2 ijms-26-03847-t002:** Primer sequences used in the study.

Genes	GenBank No.	Primers (5′-3′)
*NKCC1a*	XM_038733157.1	F: GATGATGGCAAAGCTCCAACT R: TGTGCCCTTCCCTTGTTTCTT
*β-action*	XM_020651307.1	F: AAAGGGAAATCGTGCGTGAC R: AAGGAAGGCTGGAAGAGGG

## Data Availability

Data is contained within the article.

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
