# Peer review of "Osmoregulation and Physiological Response of Largemouth Bass (Micropterus salmoides) Juvenile to Different Salinity Stresses"

_ijms, 2025, doi:10.3390/ijms26083847_

Round 1
Reviewer 1 Report
Comments and Suggestions for Authors
This manuscript shows some interesting results and maybe helpful for certain readerships.
Some comments for author to consider to make it better
1) The experimental design and procedures should be clear and in details as much as possible, with proper references to ensure its repeatability
2) The Figure 1B seems not correct, not at least not clearly presented with a potential to mislead others. please double check to make it clear
3) The rationale of 96 h should be explained or justified. The method to determine Median-Lethal Concentration should comply with related standards or methods
4) Figure 2 and Figure 3 did not show obvious differences to make it well distinguished with others
5) The discussion section is lean and should give more references (related) to support or protrude the novelty or key findings in this study.
6) In the conclusion section, the significance and novelty should be highlighted to attract broad readerships or provide insightful opinions for practical applications.
Comments on the Quality of English LanguageProfessional English polishing is advised to make it concise and well reading.
Reviewer 2 Report
Comments and Suggestions for Authors
Major remarks
- Explain the availability of this manuscript in its unchanged form on the following website: https://papers.ssrn.com/sol3/papers.cfm?abstract_id=5052334
- In the reference list, a review of older literature is provided. I recommend including a review of recent literature from the last five years.
- Avoid grouping the references in the Introduction. I recommend discussing each reference individually.
Minor remarks
- Avoid the use of the first-person plural. Only the third-person singular is acceptable in the scientific paper. For this reason, I recommend modifying all of these sentences.
- Line 58: Insert a full stop after Reference [13].
- Line 91: Provide a blank space between '96' and 'h', and also, '50' should be presented in subscript in '96h-LCâ‚…â‚€'.
- In the manuscript, the blank spaces should be provided between quantities and units.
- Figure 1: The figure title should clearly define the figures under A and B.
- Lines 102, 105, 142, 143, etc.: Delete a full stop after Figure.
- In Figures 2 and 3, what type of figures are presented and at which magnification? Are they microscopic?
- In the title of Figures 5-7, the different letters should also be explained.
- Instead of ’hours’, use an adequate unit. In this case, use ’h’.
- Lines 176, 177: Siniperca chuatsi, Ctenopharyngodon Idella, and Nile tilapia should be depicted in italics. Please carefully check the Latin terms in the manuscript, including the References list, and present them in italics.
- In section 4.7. Statistical Analysis, the software used for statistical analysis should be provided.
- Line 378: Abbreviations should be defined in this section.
Reviewer 3 Report
Comments and Suggestions for Authors
The manuscript "Osmoregulation and physiological response of largemouth bass (Micropterus salmoides) juvenile to different salinity stresses" investigates the osmoregulatory, biochemical, and physiological responses of juvenile largemouth bass exposed to varying salinity levels. This study is relevant; however, some remarks must be addressed to enhance it.
1) In the abstract session, include the practical implications of the findings about selective breeding or aquaculture strategies and describe the relevant qualitative or quantitative results.
2) Mention some physiological and molecular studies regarding the largemouth bass species under salinity stress and a condensed general background on water scarcity and environmental causes.
3) Thus, include at least five of this study's most significant findings in the introduction or conclusion sections. Focusing on physiological, biochemical, and molecular responses observed under different salinity conditions and explaining their relevance to osmoregulation and aquaculture applications.
4) The authors must justify why the selected salinity levels and exposure durations are ecologically or commercially relevant. Provide more details on statistical tests, assumptions, and software used for data analysis.
5) Figures 2 and 3 lack resolution and are hard to understand. They also need to be explained better. By the way, what does each letter in the picture mean?
6) Some results are sometimes repeated in the text without a more profound analysis.
7) No detailed explanation of the application of one-way ANOVA with multiple comparisons was reportedly used to determine significant differences between the control and treatment groups. The manuscript should clearly describe the statistical workflow, including how assumptions for ANOVA were tested.
8) Please specify the statistical software and version used for the analysis. Additionally, report p-values or significance markers obtained in this study.
Round 2
Reviewer 2 Report
Comments and Suggestions for Authors
The manuscript is well prepared according to the previously given recommendations, but I have some ambiguities.
Can you explain why this manuscript is already available on the following website: https://papers.ssrn.com/sol3/papers.cfm?abstract_id=5052334? I am not sure what this site represents. Is it the journal website?
Author Response
Dear reviewer
I sincerely appreciate your attention to this matter and would like to provide a clear explanation.
When I first submitted my manuscript to another journal, for research promotion and peer exchange, I chose the “Agree to publish in preprint” option, which automatically published the draft manuscript on the public website of the preprint. Although the manuscript was rejected by another journal, the preprint remained published. Nowadays, with the rapid development of the scientific field, more and more studies choose to publish preprints before publication for the sake of timeliness. Moreover, it is also generally accepted that the publication of a preprint is not the same as a formal publication. I fully understand the importance of originality in academic publishing and confirm that this manuscript has never been formally published elsewhere. I would be grateful if you could consider this explanation and allow the review process to proceed. Please let me know if any additional clarification or documentation is required.
Thank you for your understanding.
Sincerely,
Yang Liu
15731986210@163.com
Reviewer 3 Report
Comments and Suggestions for Authors
I want to mention that the authors have made significant revisions to the manuscript titled 'Osmoregulation and Physiological Response of Largemouth Bass (Micropterus salmoides) Juvenile to Different Salinity Stresses,' where all previous concerns and recommendations have been carefully considered and effectively resolved in the revised version. Therefore, I recommend publishing this manuscript in its present form.
Author Response
Thank you very much for the valuable comments and recognition on the manuscript.
Round 3
Reviewer 2 Report
Comments and Suggestions for Authors
The manuscript can be accepted for publication in the present form.